# Adipokines as Clinically Relevant Therapeutic Targets in Obesity

**DOI:** 10.3390/biomedicines11051427

**Published:** 2023-05-11

**Authors:** Marleen Würfel, Matthias Blüher, Michael Stumvoll, Thomas Ebert, Peter Kovacs, Anke Tönjes, Jana Breitfeld

**Affiliations:** 1Department of Medicine III, Division of Endocrinology, Nephrology and Rheumatology, University of Leipzig, Liebigstr. 18, 04103 Leipzig, Germany; marleen.wuerfel@medizin.uni-leipzig.de (M.W.);; 2Helmholtz Institute for Metabolic, Obesity and Vascular Research (HI-MAG), Helmholtz Center Munich at the University of Leipzig and the University of Leipzig Medical Center, 04103 Leipzig, Germany; 3German Center for Diabetes Research (DZD), 85764 Neuherberg, Germany

**Keywords:** adipokines, obesity, biomarker, therapeutic strategy

## Abstract

Adipokines provide an outstanding role in the comprehensive etiology of obesity and may link adipose tissue dysfunction to further metabolic and cardiovascular complications. Although several adipokines have been identified in terms of their physiological roles, many regulatory circuits remain unclear and translation from experimental studies to clinical applications has yet to occur. Nevertheless, due to their complex metabolic properties, adipokines offer immense potential for their use both as obesity-associated biomarkers and as relevant treatment strategies for overweight, obesity and metabolic comorbidities. To provide an overview of the current clinical use of adipokines, this review summarizes clinical studies investigating the potential of various adipokines with respect to diagnostic and therapeutic treatment strategies for obesity and linked metabolic disorders. Furthermore, an overview of adipokines, for which a potential for clinical use has been demonstrated in experimental studies to date, will be presented. In particular, promising data revealed that fibroblast growth factor (FGF)-19, FGF-21 and leptin offer great potential for future clinical application in the treatment of obesity and related comorbidities. Based on data from animal studies or other clinical applications in addition to obesity, adipokines including adiponectin, vaspin, resistin, chemerin, visfatin, bone morphogenetic protein 7 (BMP-7) and tumor necrosis factor alpha (TNF-α) provide potential for human clinical application.

## 1. Introduction

Adipose tissue (AT) acts as an endocrine organ producing more than 600 secretory proteins, the so-called adipo(cyto)kines, which are involved in a variety of biological processes [1]. Impaired adipokine secretion is associated with AT dysfunction in obesity, which is characterized by adipocyte hypertrophy and hyperplasia, inflammatory conditions, extracellular matrix remodelling and fibrosis [2].

Adipokines as secreted by white AT (WAT), can be described as “molecular messengers” which regulate among other processes the energy balance of the body [3]. They can act in an autocrine, paracrine or endocrine manner and interact with various organs, e.g., the brain, vasculature, muscle, pancreas and liver [4]. Further central actions include the regulation of appetite and eating behavior, insulin secretion, inflammation, lipid metabolism, blood pressure or reproduction [4,5,6]. Nevertheless, in humans with obesity, adipokine expression patterns alter towards an increase in expression of pro-inflammatory adipokines and a down-regulation of anti-inflammatory adipokines resulting in the development of obesity-associated metabolic disorders, such as insulin resistance (IR), dyslipidemia, weight gain and vascular dysfunction [7,8]. Due to their diverse mechanisms of action and effects on various metabolic regulatory circuits, adipokines offer a potential use as biomarkers for early detection of metabolic diseases or deployment as an effective therapeutic option. However, even now, many metabolic mechanisms and actions of adipokines still remain unclear. In particular, inconsistency of data due to studies with small cohorts, different ethnicities, and conflicting study results currently represent a missing link between experimental studies and the translation of their data into clinical application. With respect to their multifaceted regulatory circuits, adipokines could also provide targets as obesity-associated biomarkers. This potential is based on data from various studies linking adipokine serum concentrations with adverse metabolic states. For example, concentrations of adiponectin showed lower levels in states of obesity and type 2 diabetes (T2D) [9,10], as did concentrations of chemerin, which were positively correlated with higher body mass index (BMI) levels [11,12]. Defining measurable biomarkers that reflect the current metabolic state, and thus, characterize increased risk for the development of obesity and relevant comorbidities could be a valuable tool to detect early states of AT inflammation and dysfunction and prevent the magnitude of widespread metabolic derangements. Altogether, adipokines do not only render an eminent potential for diagnostic and prognostic purposes, but also for new pharmacological treatments.

The purpose of this work is to review the current knowledge on adipokines with respect to their potential as biomarkers or therapeutic agents in obesity and its metabolic comorbidities.

## 2. Adipokines with Approval for Application in Humans

### 2.1. FGF-21 (Fibroblast Growth Factor-21)

To date, several studies identified FGF-21 as an important player in the pathogenesis of various cardiovascular and metabolic diseases, such as atherosclerosis, coronary heart disease, myocardial infarction, obesity or non-alcoholic fatty liver disease (NAFLD) [13]. FGF-21 acts in the liver, AT, brain, pancreas, heart and kidneys via its co-factor ß-Klotho (KLB), as well as its receptors (FGFR) [14]. The liver is described as one of the major targets of FGF-21 under physiological circumstances [15]. In states of nutritional deficit, transcription factors, such as peroxisome proliferator-activated receptor (PPAR)-α or activating transcription factor 4 (ATF4), induce FGF-21 expression [16,17]. Contrariwise, in nutritional excess, FGF-21 expression is promoted by carbohydrate response element-binding protein (ChREBP) for the regulation of lipogenesis [18]. Generally, in the liver, FGF-21 is supposed to regulate lipid and fatty acid metabolism, whereas in AT, FGF-21 regulates lipolysis, glucose metabolism and thermogenesis (reviewed in [13]). To modulate lipid and glucose metabolism, FGF-21 targets signalling pathways such as AMPK, SIRT1 and molecules related to the gut microbiota. In detail, FGF-21 is supposed to exert antioxidative and lipid lowering effects via the FGF-21-calcium/calmodulin-dependent protein kinase AMPKα pathway in endothelial cells [19], the FGF-21-LKB1-AMPK-ACC1 pathway in macrophages [20] or via the FGF-21-AMPK-Akt2-Nrf2 or FGF-21-AMPK-ACC pathway in cardiomyocytes [21]. By mediating AMP-related pathways, FGF-21 has a positive effect on the progression of cardiovascular diseases and other metabolic disorders. In addition, FGF-21 is described as mediating reduced inflammatory oxidative stress in cardiovascular diseases (CVD) by targeting SIRT1 [22]. Moreover, gut microbiota is revealed as one target of FGF-21’s actions. Exemplarily, FGF-21 is supposed to stimulate hepatic β-oxidation, leading to the production of β-hydroxybutyrate, which therefore influences gut inflammation by inhibiting nod-like receptor family pyrin domain containing 3 (NLRP3) [23].

By showing a protective role in lipid, glucose, and energy homeostasis [23], FGF-21 represents an extremely promising therapeutic target in the treatment of obesity and T2D. It stimulates the release of adiponectin and is expressed in adipocytes, pancreatic acinar cells and in the nervous system of the mouse [24,25]. However, the native FGF-21 is unusable for clinical application because of poor biophysical and pharmacokinetic features [26]. For this reason, diverse FGF-21 analogues and mimetics or agonistic monoclonal antibodies for the FGFR1-ß-klotho receptor complexes have been developed. Several of them have already been included in early phases of clinical studies with patients with obesity, T2D or non-alcoholic steatohepatitis and showed promising results in improving various metabolic states. Described as anti-obesity and anti-diabetic, FGF-21 and its receptor agonists improve insulin sensitivity (IS) and promote weight loss by stimulating brown fat thermogenesis and increasing release of adiponectin serum levels [27,28]. Nevertheless, several studies reported paradoxically elevated circulating FGF-21 levels in patients with increased BMI and T2D [29,30]. Therefore, on the one hand a possible resistance to beneficial FGF-21-mediated effects in states of metabolic disorders, such as obesity or T2D, and on the other hand a reactive increase to overcome the increased energy income and triglyceride (TG) accumulation is discussed (reviewed in [31]). To explain the increased FGF-21 levels in humans two possible mechanisms have been proposed. The first one outlines Fibroblast Activation Protein (FAP) as a serine protease cleaving and inactivating FGF-21, thus limiting the amount of active FGF-21 and restricting its protective metabolic effects [32]. The second involves a single-nucleotide polymorphism (SNP) in the *KLB*-gene described to be associated with reduced KLB levels in AT and, therefore, higher BMI in obese patients [33]. Since KLB is known to be a co-receptor of FGF-21, its amount may have a direct effect on FGF-21 action. While endogenous FGF-21 appears to be ineffective at higher circulating FGF-21 levels, high pharmacological doses of FGF-21 analogues, agonistic antibodies or proteins targeting KLB could exert its effects and contribute to body weight reduction, glucose tolerance and lower circulating lipid levels [27,34,35]. One explanation might be, that the expression of KLB and FGF-21 receptor 1 and 3 in the liver is increased in patients with obesity, which might result in an increased responsiveness to exogenous FGF-21 [31].

To evaluate its clinical relevance and to substantiate pharmacologic effects of exogenous FGF-21, the long-acting FGF-21 analogue PF-05231023 (a fusion of human FGF-21 to human IgG), was administered to obese cynomolgus monkeys which firstly responded with a decrease in food intake and reduced body weight [35]. To reproduce these effects in humans, administration of PF-05231023 was tested in a placebo-controlled, multiple ascending-dose study in overweight/obese subjects with T2D, which recapitulated many of the metabolic effects observed in preclinical species. Briefly, PF-05231023 was tested intravenously (i.v.) in a phase 1b study in overweight/obese subjects with T2D already treated with metformin for four weeks. Significant reductions in body weight accompanied with lower total cholesterol, low-density-lipoprotein (LDL-C) and increased high-density-lipoprotein (HDL-C) were observed for the group receiving epp compared with placebo. FGF-21 analogue treatment also increased circulating adiponectin levels, as already shown in previous animal [36], as well as human, studies [34]. The results support previous findings of Gaich et al., who utilized the FGF-21 analogue LY2405319 in a randomized, placebo-controlled double-blind proof-of-concept trial in patients with obesity and T2D and supplied the first proof-of-concept example for the biological potential of FGF-21 in humans [34]. Thus, a statistically significant dose-dependent increase in HDL-C and circulating adiponectin levels, but a decrease in circulating triglyceride levels as well as reductions in mean body weight, could be detected.

Another first-in-human phase 1 trial to asses underlining mechanisms of weight loss after FGF-21 analogue treatment established the administration of a single-dose of the bispecific anti-FGFR1/KLB agonist antibody BFKB8488 [37]. Here, a subcutaneous (s.c.) injection caused body weight reduction and improved cardiometabolic parameters, such as increased HDL-C and total serum adiponectin, but decreased LDL-C, TG and fasting insulin [37]. This was the first study investigating eating behaviour and FGF-21 levels in humans, noting a direction of reduced preferences for sweets and carbohydrate intake after administration. Moreover, as reason for a significant weight loss, an increase in satiety and a reduced caloric intake with preferred protein or fat than carbohydrates was suggested [37]. Recently, Epperlein et al. investigated the connection between FGF-21 levels and food cravings and eating behaviour types [38] and showed negative correlations between FGF-21 levels and “disinhibition”, which defines a loss of controlling food intake. Although causal mechanisms remain to be explored, the study clearly suggested an involvement of FGF-21 in eating behaviour, which was further strengthened by independent works reporting altered food preferences after FGF-21 administration. Of note, several nutritional signals, such as high carbohydrate and alcohol intake, fat intake, low protein intake or fasting, are supposed to induce the hepatic FGF-21 expression in the liver [39,40,41]. In fact, FGF-21 implies an outstanding role as a liver-derived hormone regulating macronutrient intake and energy homeostasis by directly signalling to the brain to maintain energy homeostasis. However, evidence for physiologic circuits of FGF-21 in the human brain are still missing, although it was shown that FGF-21 and its co-receptor KLB are both expressed in the central nervous system (CNS), as already known from mouse studies [42]. Other researchers emphasize the importance of increasing endogenously circulating FGF-21, as studies showed that transgenic mice overexpressing FGF-21 were resistant to diet-induced weight gain [43]. Following the identification of FAP as the FGF-21-cleaving protein [44] and PPAR-α as a transcriptional regulator [16], there may be therapeutic potential to increase endogenous FGF-21 levels by stimulating PPAR-α and inhibiting FAP. With the fact that FAP has an amino acid identity to dipeptidyl peptidase IV (DPPIV), which is one target of anti-diabetic drugs, stabilizing the endogenous glucagon-like peptide-1 (GLP-1) could increase the activity of circulating FGF-21 levels in humans [45]. The mechanism could be similar to the DPPIV-mediated effect and in accordance with the resulting clinical success in elevating functional levels of FGF-21 by reducing inhibitory mechanisms.

Interestingly, the therapeutic benefit of FGF-21 has also attracted attention in clinical trials in patients with NAFLD. Specifically, the FGF-21 analogue pegbelfermin (BMS-986036), has been shown to improve not only metabolic markers, but also liver fibrosis in patients with obesity and T2D [46]. In a phase 2a study, obese patients with biopsy-proven NAFLD and a hepatic fat percentage of at least 10% were enrolled and received 16 weeks of s.c. injections of either placebo or pegbelfermin. After the treatment period, a significant decrease in absolute liver fat percentage was observed in patients with non-alcoholic steatohepatitis (NASH) in the group receiving pegbelfermin compared to placebo-treated participants [46]. In addition, the use of pegbelfermin decreased serum concentrations of secondary bile acids [47]. Hence, a possible role of this analogue on the modulation of bile acid synthesis, and thus, its potential use as a treatment modality for fatty liver, was suggested [47]. Another randomized, double-blind, placebo-controlled phase 2 trial in patients with T2D and obesity presented significant improved HDL-C and TG, fibrosis biomarkers, as well as increased adiponectin levels after administration of s.c. pegbelfermin for 12 weeks [48]. To evaluate its safety and efficacy, pegbelfermin is currently under investigation in phase 2b trials in adults suffering from NASH and liver fibrosis (NCT03486899), as well as in adults with NASH and compensated liver cirrhosis (NCT03486912), but more studies are needed to assess long term effects.

Consistent with the addressed therapeutic potential of FGF-21 in the treatment of NAFLD, the FcFGF21 analogue efruxifermin (AKR-001) has also received scientific attention. Initially, a phase 1 study of efruxifermin at multiple ascending doses was conducted in patients with T2D to evaluate safety, tolerability, and pharmacodynamics and kinetics after a four-week treatment period [49]. After the treatment period, significant reductions in plasma TG and LDL-C, as well as increased HDL-C and adiponectin levels, could be detected. Recently, new data presented the clinical profile of efruxifermin in terms of its efficacy and safety within a 24-week phase 2b study in patients with pre-cirrhotic NASH, fibrosis stage 2 or 3 (F2–F3) [50]. After treatment over 12 weeks, hepatic steatosis could be reduced in 70% of the patients. Likewise, a raise in circulating adiponectin could be measured, as well as improvements in body weight, glycaemic control and liver fat.

As another genetically engineered variant of endogenous FGF-21, LLF580 was investigated in a double-blind multicentre trial in adults with obesity and hypertriglyceridemia [51]. After a treatment period of four weeks, probands receiving LLF580 presented lower levels of TG, LDL-C, IR and liver fat, but higher HDL-C and adiponectin levels compared to those receiving placebo. Moreover, pegozafermin (BIO89-100) as a glycopegylated FGF-21 analogue was tested in a randomized, double-blind, placebo-controlled study with participants suffering from NASH or NAFLD and a high risk of NASH [52]. Compared to the placebo, pegozafermin significantly reduced hepatic fat fraction, improved triglyceride levels, LDL-C, HDL-C, adiponectin and body weight, among others.

Through the improved, but still insufficient understanding of its influence on circulating lipid profile and body weight regulation, FGF-21 has become an attractive player in clinical investigations (Table 1). However, its role in the control of glucose homeostasis has not been fully elucidated, which opens up room for further investigation. Further clinical trials are needed to identify potential FGF-21 molecules with all its benefits or limitations in humans. Further, to improve efficacy and safety of FGF-21-based therapeutic strategies and with regard to a possible AT selective FGF-21 resistance in states of obesity, the development of tissue specific analogues has to be discussed.

### 2.2. FGF-19 (Fibroblast Growth Factor-19)

FGF-19 is another protein of the FGF family and plays an important physiological role in the regulation of glucose and lipid metabolism, thus offering promising approaches for therapeutic strategies in metabolic disorders [53]. As an intestinal hormone with the origin in the distal small intestine, FGF-19 signals to the liver by the portal circulation in a diurnal pattern (reviewed in [54]). Moreover, its expression is regulated by the farnesoid-X-receptor (FXR), that regulates bile acid biosynthesis and transport, so that FGF-19 is defined as an important coordinator of bile acid biosynthesis and gallbladder filling [54]. Initially, FGF-19 is secreted in response to nutritional states resulting in increased postprandial concentrations and examining FGF-19 as an insulin-like hormone [55].

In patients with T2D, FGF-19 serum levels are inversely correlated with markers of the metabolic syndrome (MS), such as BMI, TGs and glycated haemoglobin A1c (HbA1c) [56]. Furthermore, FGF-19 knockout mice on a normal chow diet developed adiposity, glucose intolerance, IR and hyperlipidemia [57]. On the contrary, body weight loss and increased energy expenditure prevailed in transgenic mice expressing human FGF-19 and these mice furthermore presented a resistance to obesity and insulin desensitization [58]. Preclinical research in rats further presented that intracerebroventricular (i.c.v.) administration increased energy expenditure and lowered serum insulin, leptin, TG, cholesterol and food intake over 24 h, but also induced weight loss and improved glucose tolerance [58,59]. In contrast, an inhibition of the CNS FGFRs led to an increased food intake and an impaired glucose tolerance [59].

First results from application of pharmacological agents that act on FGF-19 expression, such as aldafermin (NGM282), show highly promising results (reviewed in [60]). As a non-mitogenic FGF-19 variant, NGM282 was applied to biopsy-confirmed patients with NASH in a randomised, double-blind, placebo-controlled phase 2 trial [61]. Participants received placebo or subcutaneous NGM282 for a treatment period of 12 weeks. Patients receiving NGM282 showed a relevant reduction in absolute liver fat content compared to baseline values [61]. Likewise, metabolic markers related to the amount of liver damage were also significantly reduced. So far, treatment with NGM282 went along with an acceptable safety profile in the investigated participants. Recently, data of the 24-week randomized, double-blind, placebo-controlled multi-centre phase 2b trial (ALLPINE 2/3) presented outcomes about the efficacy, safety and tolerability of NGM282 in patients with histologically confirmed NASH [62]. After daily doses of NGM282, significant improvements in liver fibrosis without worsening of NASH was observed, while aldafermin was generally well tolerated. Further, in a multi-centre, randomised, double-blind placebo controlled trial, patients with primary biliary cholangitis (PBC) received s.c. daily doses of NGM282 or placebo for 28 days [63]. The levels of alkaline phosphatase (ALP) and transaminase, both correlating with higher risk of death or the need of liver transplantation in patients with PBC, showed a significant decrease after NGM282 treatment [63] (Table 1).

### 2.3. Leptin

Leptin has obtained considerable attention as a relevant metabolic player in the regulation of central and peripheral glucose and lipid metabolism through lowering blood glucose levels and exerting anti-lipogenic effects, targeting endocrine pancreas, liver, AT, immune cells, skeletal muscle and the cardiovascular system [64]. Moreover, leptin was extensively studied regarding its regulation of appetite by acting in the CNS. Indeed, it is supposed to induce anorectic mechanisms by inhibiting neuropeptide Y (NPY)- and Agouti-related peptide (AGRP)-expressing neurons, but activating proopiomelanocortin (POMC) of the arcuate nucleus (ARC) [65]. Therefore, leptin maintains body weight homeostasis via a negative feedback loop between AT and the hypothalamus: increasing numbers of fat cells lead to increased leptin levels binding to leptin receptors (LEP-R) in the brain, which in turn inhibit food intake [66]. Leptin is also expressed in mesolimbic dopamine neurons, that in turn involve processes of motivation and reward [67]. In addition, it is acknowledged that leptin impacts β-cell function by stimulating β-cell proliferation, but suppressing cell apoptosis, as well as insulin secretion [68].

The central role in regulation of human body weight has been explicitly demonstrated in leptin-deficient individuals. Although congenital leptin deficiency is a rare disease, affected subjects suffer from obesity at a young age, accompanied by impairment of immune and reproductive systems [69,70]. Consequently, treatment with exogenous leptin resulted in decreased body weight and insulin levels, as well as in an improved lipid profile [71,72]. Nevertheless, despite promising outcomes regarding body weight loss in patients with mutations in the *leptin* gene and undergoing leptin-based therapy, especially in hyperleptemic obese people, such an effect is rarely detectable. This is most likely due to the fact that states of hyperleptinemia in overweight and obesity are associated with leptin resistance resulting in reduced energy expenditure, hyperphagia, hyperinsulinemia, hyperlipidemia, IR or diabetes [73,74].

However, current research efforts focus on various combination therapies of leptin with leptin sensitizers to overcome states of leptin resistance in obesity and to improve the metabolic state. The mechanism of leptin resistance appears to be a vicious cycle: in states of obesity and hyperleptinemia, impaired leptin-mediated effects in turn promote further weight gain and prevent an efficient therapeutic use of exogenously supplied leptin, which is unable to exert its anorexic effects [75,76]. One of the first human studies investigating a responsiveness of exogenous s.c. leptin administration using recombinant methionyl human leptin (r-metHuLeptin) presented relevant weight and fat mass loss in patients with obesity and elevated endogenous serum leptin concentrations after an initial treatment period of four weeks [77]. The authors proposed an elevation of centrally acting leptin concentrations as one of the underlying mechanisms of body weight loss, since in animal studies, a direct central administration of recombinant leptin induced weight loss through decreased food consumption [78]. Further studies with leptin-based treatment strategies presented conflicting results regarding weight reduction in individuals with obesity. Several studies failed to replicate beneficial effects on body weight in patients treated with recombinant leptin in combination with a caloric diet [79,80]. In hyperleptinemic patients with T2D, a 16-weeks-long treatment with r-metHuLeptin/metreleptin reduced HbA1c, but neither reduced body weight nor circulating inflammatory markers [81]. Moreover, levels of total leptin, leptin-binding protein and anti-leptin antibody increased, thereby limiting free leptin availability. In line with this, in a further study, two randomized controlled trials were conducted in patients with obesity and treated with metreleptin and pramlintide for 20–52 weeks [82]. Likewise, patients who had lipodystrophy were treated with metreleptin for two months to 12 years. After 4–6 months of treatment in most patients, anti-metreleptin antibodies developed, which might have caused the loss of efficiency in reducing body weight [82] (Table 1).

In the last decade, treatment with exogenous leptin has gained more attention in the therapeutic management of lipodystrophy. Lipodystrophy comprises a heterogeneous and rare complex of congenital or acquired syndromes characterized by a lack of s.c. AT without nutrient deprivation, manifested throughout the whole body or only in specific regions [83]. In addition to severe metabolic complications, such as IR and dyslipidaemia in the form of severe hypertriglyceridemia and low HDL-C [84], the loss of s.c. AT also leads to low circulating levels of the adipokine leptin [85]. Therefore, several clinical trials tested the effectiveness of exogenous leptin administration to patients suffering from lipodystrophy. As a result, a study has shown that metreleptin administration in female patients with lipodystrophy leads to an increased satiety time and a decrease in circulating ghrelin levels, supporting the important role of leptin in the regulation of human appetite [86]. Moreover, HbA1c, TG and LDL-C, as well as albuminuria, decreased in patients receiving metreleptin [87,88]. Its potential in neuroendocrine control was highlighted by Schlögl et al. who investigated the long term effects of metreleptin treatment on human eating behaviour and brain connectivity in leptin deficient patients [67]. Specifically, the clinical trial showed that long-term treatment with metreleptin in drug-naïve lipodystrophic patients resulted in changes in the central nervous network and decreased the feeling of hunger with longer lasting feeling of satiety. Additionally, the results suggest an improvement in homeostatic satiety signalling via the hypothalamus as one control centre of human hunger regulation. These promising findings of a beneficial influence of external leptin on appetite regulation were verified by a further study investigating the effect of metreleptin administration with regard to neurobehavioral changes [89]. Consistent with the previous results, the long-term administration also increased satiety measured on a visual analogue scale, while the feeling of hunger was subjectively reduced. Data from a study of the relationship between eating behaviour, body weight and various adipokine concentrations found that higher serum leptin levels correlated positively with higher levels of the eating behaviour trait “disinhibition” [90]. Constructed mediation analyses further hypothesised that the positive association between leptin and higher BMI levels could be mediated by human eating behaviour, in that higher leptin concentrations lead to disinhibited eating, and thus, higher BMI levels [90] (Table 1).

When discussing the potential of leptin-based pharmacotherapy for overweight and obesity it is important to consider that the inconsistent data were often driven by differences in populations studied, the study design and the regimen of the therapy, including dosing, the pharmacological structure of leptin and the trials’ inclusion criteria. Relevant factors influencing therapeutic success also involve the amount of circulating blood leptin, as well as mechanisms leading to “leptin resistance”, and thus, to the loss of leptin-mediated protective effects. Therefore, focusing on a combination of leptin with its sensitizers is inevitable to increase the treatment’s efficacy in future clinical trials. Exemplarily, previous studies detected that inhibition of cytokine signalling 3 (SOCS3) and tyrosine phosphatase 1B (PTP1B)-suppressors as negative regulators of leptin signalling pathway lead to improved metabolic outcomes by increasing the sensitivity of leptin treatment [91,92]. In addition, withaferin A has been identified as a sensitizer of endogenous leptin. After treatment of diet-induced obese mice with withaferin A, a significant reduction of body weight, as well as improved hepatic steatosis and insulin sensitivity, was detected [93]. Likewise, celastrol was described as a leptin sensitizer, which reduced food intake and body weight in hyperleptinemic diet-induced obese mice [94]. Both sensitizers seem to affect endoplasmatic reticulum (ER) stress-signalling pathways [93,94].

In addition to the efforts focusing on identifying potential inhibitors and activators of the leptin receptor or transcripts in the leptin signalling pathway, some studies also addressed the theory of therapeutic lowering of leptin levels in patients with obesity and consequent hyperleptinemia. This could ensure a renewed sensitization to endogenous leptin, leading to weight loss and improved metabolic parameters. To support this hypothesis, Zhao et al. presented significant weight loss and improved glucose tolerance in initially hyperleptinemic mice after doxycycline-induced lowering of AT leptin levels [95]. These findings proved that hyperleptinaemia in obesity led to further metabolic disturbances, as a partial reduction of circulating leptin suppressed an increase in body weight and associated metabolic disturbances in mice. The same results could be replicated by using human antibodies against circulating leptin in obese mice [95]. In addition, use of the antibodies resulted in reduced food intake mediated by increased thermogenesis in AT, suggesting the efficacy of restored leptin sensitivity. Further results showed that neutralizing antibodies restored leptin action in POMC neurons expressing leptin receptors and consequently significantly affected food intake [95].

Taken together, the results of recent studies show immense research potential for the clinical use of leptin in the treatment of obesity (Table 1). However, in order to test new scientific approaches in human clinical trials breaking through the putative resistance of endogenous leptin in obesity, the mechanisms of action and signalling pathways of leptin remain to be fully determined.

## 3. Selected Adipokines with Potential Clinical Relevance

### 3.1. Adiponectin

As another adipocyte-derived hormone, adiponectin holds pleiotropic effects in regulating IS [96], but also provides anti-atherogenic [97] and anti-inflammatory habits [98]. Moreover, adiponectin is not only produced in AT, but also expressed in osteoblasts, liver, myocytes, epithelial cells or placenta (reviewed in [99]). It is already known that higher adiponectin serum levels are associated with a reduced risk for the development of T2D, obesity and obesity-related disorders (reviewed in [100]). On the contrary, in states of obesity and T2D, adiponectin concentrations are lowered [9,10].

To maintain the protective properties of adiponectin in obesity and prevent the development of T2D, obesity, and cardiovascular disease, the current therapeutic approach with the use of pharmacological interventions focuses on restoring increased AT-associated adipokine secretion. In accordance with physiological effects, adiponectin replacement therapy has attracted considerable attention in the treatment of obesity, IR or T2D. Experimental studies in obese mice showed that an s.c. application of globular adiponectin (gAcrp39) resulted in increased plasma adiponectin concentrations and consequently with ameliorated lipid-induced IR [101]. One mechanism discussed to define possible regulatory actions was the increased fatty acid oxidation in skeletal muscle through adiponectin treatment. Furthermore, not only the ectopic lipid content in liver and skeletal muscle was reduced, but also plasma glucose levels as a result of improved insulin signalling [101]. However, underlying physiological mechanisms by which the recombinant adiponectin improves lipid profile remained under debate. Previous studies outlined PPAR-γ agonists, such as the thiazolidinediones (TZDs) pioglitazone and rosiglitazone, as inducers of increased adiponectin expression in rodents and humans (reviewed in [102]). Known as anti-diabetic drugs, their insulin-sensitizing actions are mediated partly due to increased serum adiponectin levels [103]. Nevertheless, besides its potential effects in inducing adiponectin expression, PPAR-γ agonists especially presented several side effects, such as weight gain, cardiac failure or edema, suggesting an imbalance between clinical benefits of elevated adiponectin concentrations and expected unfavourable metabolic side effects [102].

Further investigations focused on increasing the expression of adiponectin receptors to improve adiponectin signalling in states of obesity. Therefore, after identification of AdipoR1 and AdipoR2 as the primary acting receptors which are downregulated in obesity-associated IR and diabetes, one study reported that overexpression of AdipoR1 in skeletal muscle of rats improved glucose uptake [104]. Likewise, pioglitazone was found to prevent decreased AdipoR1 and AdipoR2 in states of hyperglycemia, and therefore, leads to improved metabolic effects [105]. A further study investigated an orally active synthetic small-molecule AdipoR agonist “AdipoRon”, that binds to both AdipoR1 and AdipoR2 [106]. After its administration to genetically obese mice, the data demonstrated a significant improvement in insulin sensitivity, glucose tolerance, and furthermore, a healthy extension of lifespan [106,107]. As the first synthetic analogue of endogenous adiponectin, AdipoRon and its promising metabolic effects are widely acknowledged. It was also shown to exert renal protective effects as AdipoRon treatment was able to revise diabetes-related renal changes in obese mice [108]. Moreover, in recent decades, adiponectin has also been studied as a mediator linking obesity and the pathogenesis of different malignancies, such as breast, endometrial, thyreoid or ovarian cancer (reviewed in [109]). In this regard, several studies showed controversial data for different cancer types. For example, a meta-analysis revealed significantly lower circulating adiponectin concentrations in patients who later developed prostate cancer than in control subjects, suggesting adiponectin as a promising biomarker for early detection of prostate cancer [110]. In contrast, another study did not find a significant association between circulating adiponectin levels and the risk of developing prostate cancer [111]. Therefore, linking the complex signalling pathways of adiponectin to cancer development and progression with the equally necessary identification of its pro- and anti-inflammatory properties still remains a challenging task.

To conclude, and as reviewed by Padmalayam et al., the main approaches to develop adiponectin-based therapies involving the upregulation of adiponectin, include both an increase in circulating adiponectin levels and targeting adiponectin receptors or downstream signalling molecules [102]. In order to develop promising therapeutic strategies in the future exploiting the potential properties of adiponectin with targeted increase of its serum concentration, the exact signalling pathways need to be more precisely defined and selected. Since adiponectin affects many different metabolic pathways, the future challenge defines the differentiation of specific targets and their potential effectiveness in human participations, which explains the immense need for future clinical trials.

### 3.2. Vaspin

In the past two decades considerable attention has been devoted to the visceral AT-derived serine protease inhibitor (vaspin), that is mainly produced in skin, visceral AT, as well as in the hypothalamus and stomach, liver and pancreas [112,113,114]. Several studies found positive correlations between higher expression levels of human vaspin and increased BMI, IS and glucose tolerance in vivo, suggesting a regulatory role of vaspin in compensating disturbed metabolic states in obesity [112,115]. WAT has been identified as one of the target organs for vaspin and further investigations revealed normalization of mRNA levels of different genes, such as *glucose transporter-4* (*GLUT4*), *leptin*, *resistin* and *adiponectin*, after vaspin administration in diet-induced obese mice [115].

Furthermore, vaspin has been ascribed as a centrally and food intake-regulating function in that, for example, Klöting et al. found that i.c.v. application of recombinant vaspin resulted in a significant reduction in food intake in obese and lean mice [114]. Accordingly, it is suggested that the vaspin enhances hunger feeling, which therefore, results in a higher food intake [116]. Following the hypothesis that vaspin could inhibit a protease that itself exerts orexigenic and glucose-elevating effects, further research verified kallikrein 7 (KLK7) as a protease target of vaspin’s serpin inhibition mechanism [117]. Therefore, because KLK7 splits human insulin in their A- and B-chains, vaspin treatment increases insulin concentrations and improves IS in obese mice through the protection of insulin itself [117].

Since several studies have shown beneficial effects of recombinant vaspin in rodent models of obesity, vaspin has gained considerable attention for testing its physiological role in states of obesity and hyperglycemia in human clinical trials. In a clinical intervention study aimed at associating changes in circulating vaspin levels with weight changes, vaspin levels were measured in participants with obesity after participating in a weight loss program [118]. It was found that circulating vaspin levels decreased significantly with a concomitant decrease in body weight. Moreover, the decreased serum vaspin levels were associated with ameliorated IR. These results support the hypothesis that vaspin plays a compensatory role in antagonizing impaired metabolic parameters during obesity. In contrast, other studies could not detect any significant changes in measured vaspin concentrations after weight loss intervention [119,120].

Nevertheless, vaspin remains a promising adipokine whose mechanisms of action could be exploited in the future for potential approaches to the treatment of obesity. To this end, its specific regulatory circuits in metabolic pathways and eating behaviour must be identified and the potential directions of action highlighted.

### 3.3. Resistin

In recent years, the primarily adipose tissue-derived hormone resistin (that names resistance to insulin) has been attributed a significant key role in the association between visceral obesity and T2D. Previous studies showed increased levels of resistin in genetic or diet-induced obese mice [121,122], whereas after neutralization of circulating resistin by the antidiabetic drug rosiglitazone, enhanced insulin-stimulated glucose uptake was achieved [123]. In addition, in human individuals with IR, higher resistin concentration has been found compared to people with normal glucose metabolism, suggesting a BMI-dependent association between resistin and IR in subjects with T2D [124].

Since resistin is supposed to regulate glucose metabolism and is involved in obesity-induced T2D, hyperresistinemia has been declared as a biomarker for metabolic diseases [125]. Nevertheless, IR-related pathways and effects of resistin in humans still remain unclear. One meta-analysis of 15 studies confirmed that resistin levels are positively correlated with IR in patients with obesity or T2D, but only in those people presenting higher resistin levels in contrast to those with normal circulating levels [126]. A study looking at the underlying causes of IR postulated that resistin induces IR in hepatocytes by blocking insulin signal transduction pathways [127].

Similarly, metformin could diminish the effect of resistin and decrease its expression in human hepatocytes. One further study presented that the administration of recombinant human resistin (hResistin) leads to higher secretion of pro-inflammatory cytokines, tumor necrosis factor alpha (TNF-α) and Interleukin-12 (IL-12), pointing out resistins role as an inflammatory molecule in humans [128]. Because several studies detected different directions of the amount of circulating resistin in obesity, Qatanani et al. generated “humanized” resistin mice, which describes mice that express macrophage-specific human resistin, but no murine resistin [129]. After inducing a high-fat diet, those mice developed WAT inflammation resulting in increased lipolysis, as well as serum free fatty acids and IR [129]. One trial examining the correlation between circulating resistin concentrations and BMI, waist-to-hip ratio, or fat mass as markers of obesity showed a particularly strong correlation between resistin and anthropometric traits defining obesity [130]. Nevertheless, previous studies examining weight loss through behavioural programs, such as lifestyle modification or a hypocaloric diet, have not found a decrease in circulating levels of resistin [131,132].

While resistin is already known to play a significant role in glucose homeostasis in rodent models, it remains an open question whether this pathophysiology of IR is transferable to humans. In this regard, data are often inconclusive, leading to difficulties in interpreting the conflicting conclusions. Nevertheless, some studies suggest that resistin has potential as a therapeutic agent. One possible approach would be the pharmacological reduction of circulating resistin in stages of hyperresistinemia, with the putative goal of improving IR. This points to resistin as a potential mediator between obesity and diabetes. Future clinical trials with recombinant resistin need to be developed to exploit its possible therapeutic potential. However, prior to this, the comprehensive regulatory circuits of resistin with their precise direction of action in the association between obesity and IR in humans need to be defined.

### 3.4. Chemerin

While chemerin is expressed in various tissues, such as liver, pancreas and lung [133,134], and its receptor CMKLR1 has been identified predominantly in cells of the immune system [133], it is expressed primarily by mature adipocytes [11].

The biological role of chemerin has been reported in the development of obesity since studies detected positive correlations between chemerin expression levels with BMI and metabolic biomarkers, such as circulating lipids, blood pressure, IR and different inflammation markers [12,135,136,137]. Because of its described role as a proinflammatory cytokine [11], chemerin is thought to impact inflammation of AT contributing to obesity. However, these results might suggest that a reduction in circulating chemerin levels could lead to improved metabolic status.

As it is already known that chemerin levels are increased in obesity, further research outlined the regulation of adipogenesis, inflammation and vascular dysfunction along with its cell surface receptor chemR23, encoded by the *CMKLR1*-gene [138]. Patients suffering from obesity presented higher expression levels of chemR23 in adipocytes and skeletal muscle cells [139]. However, a study investigating the association between circulating chemerin levels and the cardiovascular phenotype in patients suffering from chronic kidney disease (CKD) presented that chemerin exerts a protective effect on vascular calcification in CKD by mediating signalling through its chemR23 in vascular smooth muscle cells [140].

One therapeutic strategy aimed at decreasing circulating chemerin levels examined the influence of the antidiabetic drugs metformin and pioglitazone on chemerin levels in patients with T2D. Thus, patients with newly diagnosed T2D were either treated with pioglitazone or metformin daily for three months [141]. Consequently, not only indices of glycemia or IR improved by both medications, but also chemerin concentrations were significantly lower. Nevertheless, more studies are needed to define the clinical significance of decreased chemerin concentrations along with T2D and related comorbidities.

In the past decades, chemerin has been described in a neuroendocrine axis [65,142]. Supporting this hypothesis, chemerin and its receptors were identified in various brain regions, such as the hypothalamus, hippocampus or prefrontal cortex [142]. Research presented that chronic i.c.v. injection of chemerin into rats increased food intake and body weight, whereas acute infusion resulted in an inhibition, both pointing out its potential role in the regulation of energy homeostasis. To expand the understanding of the relationship between eating behaviour and chemerin in humans, another study examined the association between chemerin, food intake, and body weight [90]. The data collected led to the assumption that chemerin could influence human eating behaviour, thus leading to an increase in BMI.

In conclusion, although the relevance of chemerin in the context of metabolic disorders is well known, the establishment of pharmacologically active substances is still lacking in clinical practice. This requires further research into its signalling pathways, as well as its influence on human eating behaviour, in order to be able to establish effective therapeutic strategies based on chemerin-associated regulatory circuits.

### 3.5. Visfatin

Visfatin, also known as a pre-B cell colony-enhancing factor [143], is predominantly expressed by visceral AT of mice and humans and exerts insulin-mimetic actions [144]. However, visfatin is also expressed in s.c. AT, skeletal muscle, liver, immune cells, brain cells and cardiomyocytes (reviewed in [145]). Due to its described role in the pathogenesis of IR with binding affinity to the insulin receptor at a distinct binding site, visfatin has attracted research attention [143]. Moreover, visfatin is supposed to reduce liver-dependent glucose release and stimulates glucose utilization in adipocytes and myocytes, exerting glucose-lowering effects [143].

In a double-blind, placebo-controlled crossover study with healthy participants receiving i.v. glucose infusions, glucose was found to increase circulating visfatin levels [146]. Contrariwise, another study presented reduced plasma visfatin levels in overweight female participants who received a 2-h oral glucose tolerance test (OGTT) [147]. The authors therefore discussed the importance of insulin and glucagon-like-peptide-1 (GLP-1) as potential rapid suppressors of visfatin, which could further explain the differences in visfatin expression between oral or i.v. glucose substitution.

In the past, several studies have presented conflicting results regarding the relationship between visfatin and IR in patients with obesity. Precisely, this raises the difficulty of using its potential metabolic properties as biomarkers for early detection of IR and related disorders [148,149,150]. One further study investigated the effects of recombinant visfatin on human leukocytes [151]. After stimulation, a dose-dependent induction of IL-1β, IL-1Ra, IL-6, IL-10 and TNF-α could be measured. These results indicated that visfatin is a pro-inflammatory adipokine that may be involved in the pathogenesis of inflammatory diseases.

Supporting these findings, Abdalla et al. reported that higher visfatin levels in patients with increased glucose concentrations may represent a regulatory mechanism to maintain glucose homeostasis [145]. However, as physiological insulin regulation is absent in these patients, insulin-dependent down-regulation of visfatin was accordingly missing and unphysiological high visfatin doses were found. These considered the release of inflammatory markers, leading to endothelial dysfunction and cellular proliferation [145].

Additionally, a further study examining the consequences of visfatin elevation in obesity found that visfatin leads to upregulation of certain factors that promote remodelling of extracellular matrix [152]. According to the authors, the fibrosis of AT triggered by visfatin could therefore be an obesity-related pathomechanism of this adipokine.

In conclusion, it can be stated that visfatin has significant influence on insulin-dependent metabolic pathways and the mentioned studies suggest that elevated concentrations in the status of overweight might exert a regulatory function to maintain metabolic balance. Nevertheless, excessive visfatin concentrations seem to lead to inflammatory states, resulting in the development of pathological metabolic states such as T2D and associated comorbidities. In order to make use of the beneficial properties of visfatin in the treatment of various diseases, future research will first be required to analyse other endocrine regulatory circuits and pathomechanisms on which it exerts an influence. In addition, the therapeutic benefit could be explored in clinical trials.

### 3.6. Bone Morphogenetic Protein 7 (BMP-7)

BMP-7 belongs to the bone morphogenetic proteins (BMPs), which are members of the transforming growth factor beta superfamily (TGF-ß), which in turn comprise the multifunctional growth factors [153]. Overall, BMPs are supposed to hold important roles in regulating haematopoiesis, as well as embryonic, heart, neuronal or cartilage development and postnatal bone formation [153,154]. Further studies to outline their role in regulating adipogenesis revealed that BMP-7 is able to induce *uncoupling protein-1 (UCP-1)* mRNA expression in brown pre-adipocytes, indicating its unique potential among other BMPs [155]. Further data showed that mice expressing adenoviral-mediated higher levels of BMP-7 had higher brown, but not white, fat mass [155]. They also showed higher whole-body energy expenditure and basal body temperature, which in turn resulted in decreased weight gain without any increase in physical activity or food intake. Accordingly, BMP-7 has received considerable attention as a potential therapeutic target in the treatment of obesity [155].

Interestingly, to further investigate the potential role of BMP-7 as a therapeutic option for obesity, a previous study in mice investigated its role in appetite regulation [156]. Townsend et al. detected a higher energy expidenture and decreased food intake in diet-induced mice treated with BMP-7, which therefore decreased body weight [156]. Since the discovery of BMP7 expression in the hypothalamus [157], the CNS has become an interesting target to study the relationships between BMP-7 expression, energy expenditure, and appetite regulation. Townsend et al. further revealed that i.c.v. administration of BMP-7 in mice results in lower food intake, which might be mediated by the central mTOR-p70S6 kinase pathway [156]. Thus, the authors also discussed BMP-7 as a possible anorectic factor regulating appetite via a leptin-independent hypothalamic pathway.

Nevertheless, clinical testing of its protective properties with respect to metabolic pathways is still lacking. To date, no studies have been established investigating either the efficacy of weight loss or overall metabolic control by recombinant BMP-7 in humans.

### 3.7. Tumor Necrosis Factor Alpha (TNF-α)

TNF-α is already known as a cytokine released from macrophages [158,159]. Since endogenous TNF-α is produced by AT [160,161], it is also discussed as an “adipokine”. Indeed, TNF-α holds pleiotropic properties including impacts on inflammation and cell survival, but also lipid and glucose metabolism [160,162]. Regarding its potential metabolic role mediating obesity and IR, TNF-α was found to be increased in rodent obesity [161], whereas neutralization of TNF-α in animals with obesity and IR resulted in higher IS [161]. Overall, previous studies presented an overexpression of TNF-α levels in rodents and humans suffering from obesity [161,163,164]. Due to adipose tissue dysfunction in states of obesity, WAT predominantly expresses proinflammatory adipokines and cytokines, such as Il-1b, Il-6, Il-12, TNF-α or interferon gamma, which contribute to the conditions of chronic systemic inflammation in obesity (reviewed in [165]). In particular, macrophages from AT are considered as the main source of pro-inflammatory cytokines such as Il-6 or TNF-α [165].

Hotamisligil et al. investigated *TNF-α* mRNA expression in human AT with respect to changes in expression after weight loss in states of obesity [163]. They firstly recognized that patients suffering from obesity, presented 2.5-fold higher *TNF-α* mRNA levels than controls with normal weight. One indication of the possible important function of TNF-α in regulating obesity-related IR was that a reduction in body weight after dietary treatment resulted not only in improved IS, but also in lower *TNF-α*-mRNA expression levels [163]. Furthermore, obese mice with a genetic mutation leading to lacking TNF-α presented significantly improved IS and lower circulating free fatty acids [166].

To further evaluate its clinical potential, treatment with human anti-TNF-α antibody has been investigated in patients with obesity suffering from non-insulin dependent diabetes mellitus. After a treatment period of four weeks along with TNF-α neutralization, no effect on IS could be detected [167]. A further study investigated the impact of TNF-α neutralization on IR via infusions with infliximab by performing a prospective, randomized double-blind placebo-controlled trial in healthy male patients suffering from obesity [168]. Whereas the inflammatory status (measured by high sensitivity C-reactive protein (hsCRP)) and fibrinogen was improved, IR or endothelial function did not change. This result is in line with other studies, investigating the impact of TNF-α neutralization via TNF-α antibodies such as infliximab or etanercept on IS in patients with obesity [169,170,171]. In this regard, Di Rocco et al. discussed the paracrine pathway of TNF-α as a possible reason for the absent effect of infliximab on IR in patients with obesity [169]. Specifically, infliximab tends to distribute in the vasculature, whereas its effective capacity to act in AT or muscle decreases.

Several preclinical studies in the past attempted to investigate the relationship between obesity and associated IR possibly mediated by TNF-α. However, the transfer from pre-clinical testing to clinical practical use as a therapeutic agent in patients with obesity has not yet been successful. Published human studies investigating the association of TNF-α antibodies with obesity and IR have so far failed to provide efficacy in favour of a significantly improved metabolic state that would justify their use in clinical practice as a therapeutic option. Nevertheless, as a cytokine with multiple mechanisms of action, TNF-α represents an attractive target for the treatment of various diseases.

Further clinical studies are needed to adequately understand and effectively use its potential for targeting and improving metabolism in humans as has already been initiated for other adipokines (as shown in Figure 1).

## 4. Conclusions and Outlook: The Evolutionary Path of Adipokines from Biomarker to Therapeutic Strategy

While it is commonly known that adipokine expression patterns are altered in states of AT dysfunction, exact mechanisms of how various adipokines contribute to obesity-related disorders still remain important subjects of investigation. Indeed, several adipokines can be assigned either as pro- or anti-inflammatory mechanisms of action, whereas some of them are also thought to exert their effects according to the current metabolic state. Therefore, it often remains challenging to answer the questions: (a) whether higher circulating serum adipokine levels in obesity represent a compensatory mechanism to overcome obesity-related imbalances such as IR, hyperlipidemia or uncontrolled food intake; (b) whether there is relevant resistance to multiple beneficial effects of adipokines in overweight and obesity; or (c) whether a rise or fall in concentrations provide a physiological response to AT dysfunction contributing to obesity-related metabolic diseases. For the latter, explanations regarding causality are often lacking: do altered adipokine levels contribute to AT dysfunction, or does initial obesity-associated AT dysfunction lead to altered adipokine secretion? Nevertheless, due to their multifaceted actions including appetite, energy metabolism, satiety or physical activity, adipokines still offer immense potential for their use as a diagnostic or even therapeutic tool in overweight and obesity (Table 1).

As outlined above, adipokines such as FGF-21 and leptin have not only been studied in detail with regard to their physiological effects in animal studies, but also with regard to their therapeutic benefits in humans. In summary, already initiated human clinical trials have yielded a good chance of translating the beneficial properties of certain adipokines as effective therapeutic agents into the clinical setting. For example, the presented data of treatment with FGF-21 agonists or FGF-19 antibodies or analogues showed the great potential for their use in metabolic diseases, such as obesity, through beneficial effects on IS, lipid levels or food intake (Table 1). Nevertheless, the exact mechanisms for how these adipokines regulate IS or food intake often remain the subject of investigations. A prime example of the transition of adipokine’s use into clinical practice is represented by leptin and the use of metreleptin in the treatment of patients with lipodystophy. Ongoing research is addressing leptin resistance, which needs to be overcome in order to also utilise leptin’s potential in the therapy of obesity. While these adipokines are already experiencing a breakthrough in terms of their evaluation in human studies, others have so far only been investigated in experimental studies. Due to their demonstrated correlations with other metabolic parameters, such as blood pressure, lipid levels, IR, inflammation markers, weight change or food intake, adipokines, such as adiponectin, vaspin, resistin, chemerin, visfatin, BMP-7 and TNF-α, offer the potential to be used as predictive biomarkers for the clinical diagnosis and prognosis assessment of obesity-related imbalances. In this respect, concentrations of resistin and chemerin have already presented positive correlations with IR, circulating lipids or blood pressure pointed to relevant therapeutic treatment considerations for lowering circulating levels in obesity. In agreement with these findings, TNF-α was identified as a positive correlator with obesity and IR, making it worthy of discussion as an obesity-associated diagnostic tool. In other studies, circulating serum levels of insulin-sensitizing adiponectin have been found to be lower in obesity, so future therapeutic tools will need to be developed to increase concentrations, and therefore, promote a healthier phenotype. In contrast, vaspin, as well as resistin, are thought to hold a compensatory role to overcome impaired metabolic states in obesity, while lower levels represent ameliorated IR, among others. However, the example of visfatin represents the need for further investigations to define the precise physiological circuits when various studies detected conflicting results regarding the associations between visfatin and obesity-related IR. Another aspect to be addressed in future studies is the influence of various adipokines on human eating behaviour. In this respect, future research has to find out to what extent the concentration of circulating adipokines in different metabolic states on a neuroendocrine axis affects food intake. Likewise, it has to be identified, how different eating behaviour patterns influence adipokine concentration, and thereby, influence and change metabolic states. To this extent, investigations revealed BMP-7 and chemerin to be promising candidates for therapeutically influencing human eating behaviour, as they have been shown to be regulators of food intake via hypothalamic regulatory circuits.

Before discussing several adipokines as biomarkers, associations between different serum levels and specific metabolic circumstances must be described in greater detail.

Moreover, the role of adipokines in overweight and obesity must be clearly defined within their contribution of obesity-associated pathophysiology. Future treatment strategies thus involve mechanisms of compensation, resistance, and increase or decrease in circulating levels due to multiple directions of adipokine-action in obesity. Due to their inter-individual expression variability, their often not well-defined physiological signalling pathways and different interpretation patterns depending on the current metabolic state show, to date, no indication for regular laboratory chemical determination of various adipokine serum levels with regard to the cost-benefit trade-off. Nevertheless, some clinical trials actually provide evidence for the potential in using adipokines as biomarkers or further treatment tools. The lack of data or translation of results from animal studies to human studies should not discourage future research projects from conducting new human studies with the use of adipokines, as the immense number of overweight patients makes it imperative to further pursue new therapeutic strategies for obesity and place them in the clinical context.

**Table 1 biomedicines-11-01427-t001:** Overview of the development of new pharmacological targets with current clinical testing in humans.

Adipokine	Pharmacological Adipokine (Substance Class)	Target Tissue Mediating Metabolic Effects	Mode of Action	Metabolic Effect in Humans
Fibroblast growth factor-21 (FGF-21)	BFKB8488 (bispecific anti-FGFR1/ ß-Klotho agonist antibody)	- adipose tissue - CNS	- activation in hypothalamic glutaminergic neurons [37]	- ↓ body weight - ↓ caloric intake - ↓ LDL-C - ↓ triglycerides - ↓ fasting insulin - ↑ HDL-C - ↑ adiponectin
	LLF580 (FGF21 analogue binding to FGFR1 and β-Klotho)	- liver - adipose tissue	- acting on a body-weight independent manner [51] - ↓ hepatic de novo lipogenesis [51] - ↑ fat oxidation [51] - ↓ of fatty acid flux from adipose tissue to the liver [51] - ↓ steatosis and lipotoxic damage [51] - action of LFF580 on triglyceride metabolism remains unclear [51]	- improved liver fat in patients suffering from NAFLD - ↓ triglycerides and hepatic fat - ↓ total cholesterol - ↓ LDL-C - ↓ insulin resistance - ↓ bone-specific alkaline phosphatase - ↓ procollagen type I N-terminal propeptide - ↓ osteocalcin - ↑ HDL-C - ↑ adiponectin
	Pegozafermin/BIO89-100 (FGF21 analogue binding to FGFR1 and β-Klotho)	- liver - adipose tissue	- under investigation [52]	- ↓ hepatic fat - ↓ body weight - ↓ LDL-C - ↓ non-HDL-C - ↓ serum triglycerides - ↑ HDL-C - ↑ adiponectin
	PF-05231023 (FGF-21 analogue binding to FGF R1 and β-Klotho)	- adipose tissue - CNS	- ↓ expression of adiponectin receptor (AdipoR) - ↓ expression of peroxisome proliferator activated receptor y (PPARy) - ↓ leptin - ↓ lipid synthesis - ↓ pro-inflammatory markers (IL1β, IFNy) - ↑ anti-inflammatory marker (IL10) [35] - regulation of food intake [35]	- ↓ body weight - ↓ total cholesterol - ↓ LDL-C - ↑ HDL-C - ↑ adiponectin
	LY2405319 (FGF-21 analogue FGFR1 and β-Klotho)	- liver - adipose tissue	- ↑ hepatic mitochondrial function - ↑ fatty acid oxidation - ↓ inflammatory signalling [172]	- ↓ triglycerides - ↓ body weight - ↑ HDL-C - ↑ adiponectin
	Pegbelfermin/ BMS-986036 (FGF-21 analogue FGFR1 and β-Klotho)	- liver - adipose tissue	- ↓ choloylglycine hydrolase gene expression - ↓ faecal secondary bile acid levels [47]	- improved metabolic parameters - ↓ absolute liver fat percentage in patients with non-alcoholic steatohepatitis - ↓ secondary bile acids - improved HDL-C - improved triglycerides - improved fibrosis biomarkers - ↑ adiponectin levels
	Efruxifermin/AKR-001 (FcFGF21 analogue FGFR1 and β-Klotho)	- liver - adipose tissue	- direct anti-fibrotic activity	- ↓ plasma triglycerides - ↓ LDL-C - ↑ HDL-C - ↑ adiponectin levels - ↓ hepatic steatosis - improved body weight - improved glycaemic control - improved liver fat
Fibroblast growth factor-19 (FGF-19)	Aldafermin/NGM282 (non-mitogenic FGF-19 variant)	- liver	- ↓ serum concentration of 7α-hydroxy-4-cholesten-3-one (surrogate marker for enzymatic activity of CYP7A1) [63]	- ↓ liver fat content in patients with NASH - improved liver fibrosis
Leptin	r-metHuLeptin (recombinant methionyl human leptin)	- CNS	- ↑ centrally acting - ↑ leptin concentrations [78]	- weight and fat mass loss in patients with obesity and elevated endogenous serum leptin concentrations
	metreleptin (recombinant human leptin analogue)	- CNS	- signalling via hypothalamus [67,89]	- ↑ satiety time and decreased ghrelin levels in patients with lipodystrophy - ↓ HbA1c - ↓ triglycerides - ↓ LDL-C - ↓ albuminuria

CNS = central nervous system; FGFR = fibroblast growth factor receptor; IL = interleukin; IFN = interferon; LDL-C = low-densisity-lipoprotein cholesterol; HDL-C = high-densitiy-lipoprotein cholesterol; NAFLD = non-alcoholic fatty liver disease; NASH = non-alcoholic steatohepatitis; HbA1c = glycated haemoglobin A1c; ↑ = increased; ↓ = decreased.

## Figures and Tables

**Figure 1 biomedicines-11-01427-f001:**
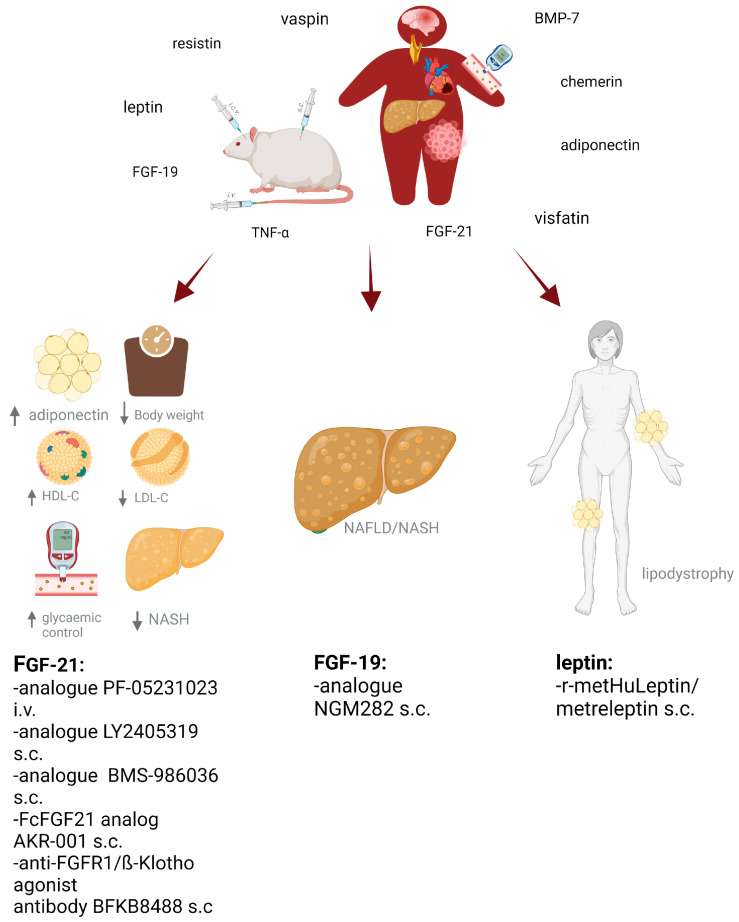
Various investigations in experimental trials with obese mice or rats examined associations between different adipokine serum levels and metabolic parameters in states of obesity-linked adipose tissue dysfunction. The adipokine secretion pattern is displaced in favour of proinflammatory cytokines and the occurrence of secondary metabolic/cardiovascular comorbidities, such as impaired insulin sensitivity, fatty liver disease, myocardial dysfunction atherosclerosis, brain dysfunction or different types of cancer. Further trials aimed at investigating effects of i.c.v, s.c. or i.v. application of adipokine agonists or antibodies on body weight, eating behaviour and different metabolic markers or liver diseases, among others. There is still a lack in translating experimental data into human clinical trials. Nevertheless, for the adipokines fibroblast growth factor (FGF)-19, FGF-21 and leptin, there exist promising transferable data for human application and their beneficial effects on obesity and metabolic disorders or diseases such as non-alcoholic fatty liver disease (NAFLD) or lipodystrophy as presented. i.v.: intravenous; s.c.: subcutaneous; i.c.v.: intracerebroventricular; LDL-C: low-density-lipoprotein; HDL-C: high-density-lipoprotein; NASH: non-alcohol steatohepatitis; NAFLD: non-alcoholic fatty liver disease, ↑ = increased; ↓ = decreased. Created with BioRender.com.

## Data Availability

Not applicable.

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
