# Peer review of "Adipokines as Clinically Relevant Therapeutic Targets in Obesity"

_biomedicines, 2023, doi:10.3390/biomedicines11051427_

Round 1

Reviewer 1 Report

The authors aim to summarize clinical studies investigating the potential of various adipokines with respect to diagnostic and therapeutic treatment strategies for obesity and linked metabolic disorders. This is an interesting review to understand the clinical use of adipokines. The topic in itself is important and of interest to the readers of biomedicines. This is a valuable submission that I recommend for publication with a few minor changes.

Minor concerns:

1.        2.1. FGF-21 (Fibroblast growth factor-21): Novel molecular targets and related pathways of FGF21 should be added in the text.

2.      3.7. Tumor necrosis factor alpha (TNF-α): The link between Pro-inflammatory cytokines (such as IL-1 and IL-6) and obesity should be added in the text.

Reviewer 2 Report

This comprehensive review focuses on the role of adipokines in obesity and related metabolic disorders. Adipokines are known to play a crucial pathogenic role in obesity, and are associated with dysfunctional adipose tissue and metabolic and cardiovascular complications. The paper provides a summary of preclinically studies investigating the potential of various adipokines for the diagnosis and treatment of obesity and related metabolic disorders, as well as an overview of the current status of adipokines used in vivo studies. Additionally, the paper outlines adipokines that have shown potential for clinical use through experimental research to date.

The review particularly highlights reports related to in vivo studies for various cytokines, including FGF-19, FGF-21, leptin, adiponectin, vaspin, resistin, chemerin, visfatin, BMP-7, and TNF-α. However, due to the large amount of information, readers may become confused. To address this issue, it would be helpful to summarize each cytokine in figures or tables, not just the last figure. Additionally, providing a summary of which cytokines (including artificial cytokines) are related to which metabolic pathways and metabolic diseases would deepen readers' understanding.

Overall, this review provides a valuable summary of the current state of knowledge regarding adipokines and their potential clinical applications for the diagnosis and treatment of obesity and related metabolic disorders.

Round 2

Reviewer 2 Report

I agree with the authors' decision to include a table in their manuscript. However, I would like to make a minor comment regarding the content of the table.

In LLF580 of Table 1, there appears to be a typo in the text 'FGF21 analogue inding to FGFR1 and β-Klotho'. Please correct this error before finalizing the manuscript.

Author Response

We thank the Reviewer for the comment regarding the typo in our manuscript. We now corrected the error and therefore finalized the manuscript.